# Dendrimers—Novel Therapeutic Approaches for Alzheimer’s Disease

**DOI:** 10.3390/biomedicines12081899

**Published:** 2024-08-20

**Authors:** Magdalena Mroziak, Gracjan Kozłowski, Weronika Kołodziejczyk, Magdalena Pszczołowska, Kamil Walczak, Jan Aleksander Beszłej, Jerzy Leszek

**Affiliations:** 1Faculty of Medicine, Wrocław Medical University, 50-367 Wrocław, Poland; 2Clinic of Psychiatry, Department of Psychiatry, Medical Department, Wrocław Medical University, 50-367 Wrocław, Poland

**Keywords:** Alzheimer’s disease, dendrimers, neurodegeneration, drug-delivery system

## Abstract

Dendrimers are covalently bonded globular nanostructures that may be used in the treatment of Alzheimer’s disease (AD). Nowadays, AD therapies are focused on improving cognitive functioning and not causal treatment. However, this may change with the use of dendrimers, which are being investigated as a drug-delivery system or as a drug per se. With their ability to inhibit amyloid formation and their anti-tau properties, they are a promising therapeutic option for AD patients. Studies have shown that dendrimers may inhibit amyloid formation in at least two ways: by blocking fibril growth and by breaking already existing fibrils. Neurofibrillary tangles (NFTs) are abnormal filaments built by tau proteins that can be accumulated in the cell, which leads to the loss of cytoskeletal microtubules and tubulin-associated proteins. Cationic phosphorus dendrimers, with their anti-tau properties, can induce the aggregation of tau into amorphous structures. Drug delivery to mitochondria is difficult due to poor transport across biological barriers, such as the inner mitochondrial membrane, which is highly negatively polarized. Dendrimers may be potential nanocarriers and increase mitochondria targeting. Another considered use of dendrimers in AD treatment is as a drug-delivery system, for example, carbamazepine (CBZ) or tacrine. They can also be used to transport siRNA into neuronal tissue and to carry antioxidants and anti-inflammatory drugs to act protectively on the nervous system.

## 1. Introduction

The number of people diagnosed with dementia is projected to triple from 47 million people to 131 million by 2050. Alzheimer’s disease (AD) is one of the main factors of dementia [1]. Currently, taking care of people afflicted with this disease is a huge challenge for the budget of developed countries. Due to the worsening of the affected people’s conditions, their needs for everyday care increase with every stage of the disease. But besides the costs of skilled nurses, hospitals, and hospices, there are also hidden costs for society, like informal caregiving. This causes a huge burden for the patient’s family and may cause emotional distress and further negative health outcomes, not only physical but also mental. Caregiving may be even more stressful in difficult times, like pandemics or war. Costs for society to care for AD patients are estimated at USD 305 billion in only one year. However, this does not include unpaid caregiving, which was evaluated at nearly USD 257 billion [2]. And this number is supposed to grow with the number of patients diagnosed with dementia [3]. This fact forces scientists to seek new options for therapy. Scientists all over the world have been carefully investigating the pathogenesis of the disease to discover new possibilities for Alzheimer’s disease treatment.

Recently, there has been interest in dendrimers. These covalently bonded globular nanostructures give hope for being a drug-delivery system for other drugs or being a drug per se. In this review, these molecules and new investigations in this topic are precisely described.

## 2. Alzheimer’s Disease—Main Facts

This is the most common neurodegenerative disease, with a prevalence of 10–30% in the population >65 years of age with an incidence of 1–3%. Most patients with Alzheimer’s disease (>95%) have a late-onset form, which means that they develop the disease after they reach 80 years old [4].

Alzheimer’s disease was named after Alois Alzheimer, a German psychiatrist and neuropathologist, who was the first to publish a neuropsychological characterization of this disease. In 1907, he precisely described 51-year-old Aguste Deter, a Frankfurt state asylum resident. After she died, he examined her brain under the microscope. Alzheimer’s observed neuritic plaques, neurofibrillary tangles, and amyloid angiopathy. Soon, this condition was known all over Europe and the United States. Due to Emil Kraepelin, his mentor, the disease started to be called Alzheimer’s.

In the second part of the past century, the methods for investigating cognitive decline have improved. With tools like the Mini-mental State Examination (MMSE) and the Montreal Cognitive Assessment (MoCA), making a diagnosis became more accurate [5]. Furthermore, to recognize the disease and monitor its course, neuroimaging methods have begun to be more and more popular. Examinations like computer tomography (CT) and magnetic resonance (MR) allow us to expose pathologies in the brain during a patient’s life [6,7]. Positron emission tomography (PET) allows the monitoring of the course of the disease and shows neurofibrillary tau aggregation [8].

## 3. Pathogenesis of Alzheimer’s Disease

The neuropathology of the disease may be characterized as the accumulation of pathological proteins, like beta-amyloid (Aβ), tau protein, and neuritic plaques (sometimes called senile plaque). Aβ plaques may be observed in people without any cognitive decline if they do not present tau pathology as well, while cored plaques are directly combined with dementia symptoms [9].

Aβ plaques usually deposit in the neocortex, then in limbic regions. Subsequently, the patient starts being symptomatic in subcortical areas and in the structures like basal ganglia and thalamus. The late stages of AD are characterized by Aβ plaques located in the midbrain, pons, and medulla oblongata [10]. Aβ may also be the reason for cerebral angiopathy, called cerebral amyloid angiopathy (CAA) [9], which may be the cause of cerebral hemorrhages [11].

Another pathological deposit in the brains of AD patients is the tau protein, which, in the correctly functioning organism, is supposed to maintain the stability of microtubules in the axons [12]. Before the symptoms occur, the pathological tau may be present in the brain [13] in a different form than in healthy patients after the processes of post-translational modifications, like acetylation [14] or hyperphosphorylation [15]. Usually, the first deposits occur in the entorhinal cortex and hippocampus [9]. Tau aggregates may create different forms, like oligomeric aggregates, straight filaments, or helical filaments [16]. Oligomeric tau is the most toxic among all types; it allows the spreading of the disease by inducing tau seeding in healthy cells [17].

Speaking about the pathogenesis of AD, it is also very important to mention mitochondrial disorders and oxidative stress. Much research has suggested that amyloid and tau deposition may be a result of mitochondrial dysfunction [18]. This is the reason why there is reduced glucose and oxygen metabolism in brains affected by AD, which was discovered with positron emission tomography (PET) and electron microscopy [19]. The decreased enzyme activity of mitochondrial enzymes in AD patients was found, especially cytochrome oxidase [20] in the frontal and cortical lobes [21]. In the study on rats, it was observed that administration of chronic sodium azide (NaN_3_) leads to partial inhibition of the mitochondrial respiratory chain by blocking cytochrome IV. It resulted in cognitive decline in rats, and histopathological examination of their brains revealed changes like nerve cell loss, thinner dendrites, corkscrew-like dendrites, and pyknotic nerve cells [22]. Similar results were obtained after rotenone application, which is a mitochondrial complex I inhibitor. Rats presented cognitive decline, and there were deposits of tau present in the cytoplasm of neuron cells [23]. Annonacin, another mitochondrial complex I inhibitor, leads to the death of neuron cells. Before this happened, there had been a redistribution of tau from axons to cell bodies [24].

Interestingly, a study conducted by Pereira revealed that the administration of Aβ to the cells resulted in inhibiting the respiratory chain, depolarizing the mitochondrial membrane, and limiting the level of oxygen consumption [25]. Furthermore, Aβ disturbs the process of mitochondrial fusion and fission [26], mitochondrial proteostasis [27], and calcium homeostasis [28].

Mitochondrial dysfunction leads to oxidative stress, which is an imbalance in the amount of antioxidants and pro-oxidants. Redox circuitry is disrupted, and increased production of reactive oxygen species (ROS) is observed. Some examples of ROS are superoxide radical anion (O_2−_), hydrogen peroxide (H_2_O_2_), hydroxyl radical (HO^−^), nitric oxide (NO), and peroxynitrite (ONOO^−^) [29]. Although ROS play an important role in defense against pathogens and take part in cellular signaling, in excessive amounts, they enhance inflammation, cellular apoptosis, and necrosis [30]. Except for mitochondria, ROS are produced in peroxisomal oxidases [31], NAD(P)H oxidases, or cytochrome P-450 enzymes [32]. Aβ stimulates the inflammatory response, which involves microglia. Microglial cells are innate immune cells in the CNS that take part in the initial pathogen defense and injury response [33]. This leads to overproduction of ROS due to Aβ binding to the mitochondrial membranes, causing incorrect energy metabolism and progressing the loss of neuron cells [34,35].

Initially, immune cells protect the brain; however, in the course of AD, the production of pro-inflammatory cytokines is initiated by glial cells. Neuroinflammation and neurotoxicity allow Aβ and tau to aggravate the brain [36]. At the beginning of AD, microglia correctly clear the brain from Aβ; however, prolonged microglia exposure leads to attenuation of the clearing mechanisms, and it continues to stimulate proinflammatory cytokines production such as NO, TNFα, IL-6, and IL-1β [37].

Interestingly, the loss of estrogens exacerbates neuroinflammation, which as well may be the reason why older women are at a higher risk for AD [38]. The administration of estrogen receptor agonists protects the brain, limiting neuroinflammation [39]. Increased levels of cortisol and activation of the hypothalamic–pituitary–adrenal (HPA) axis generated by stress is another factor that intensifies neuroinflammation [40]. Heavy metal toxicity [41], obesity, and diabetes [42] also increase neuroinflammation levels.

## 4. Treatment of Alzheimer’s Disease

Alzheimer published his famous case study only 110 years ago, and our modern understanding of the disease did not begin to accelerate until the 1980s. Since then, there has been progress in basic and translational research into treatments for Alzheimer’s disease (AD) and other dementias. In the 1980s, researchers began investigating the profiling of neuropsychological deficits associated with AD and their differentiation from other dementias. In the 1990s, previous research continued and began to identify specific cognitive mechanisms influenced by various neuropathological substrates. The 2000s focused on studying the prodromal stages of neurodegenerative disease before the onset of full dementia syndrome [43].

Recent decades have focused on advances in understanding the cellular and molecular changes associated with AD pathology. Many new and different targets for drugs were discovered. These targets include therapies to prevent the production or removal of amyloid-β (Aβ) protein, which accumulates in neuritic plaques; to prevent hyperphosphorylation and aggregation into paired helical filaments of microtubule-associated tau protein; and to keep neurons alive and functioning physiologically [44].

Two commonly used classes of drugs approved for the treatment of AD are cholinesterase enzyme inhibitors (of natural origin and synthetic and hybrid analogs) and N-methyl-d-aspartate (NMDA) antagonists. In AD, pathological processes destroy ACh-producing cells, reducing cholinergic transmission through the brain. The mechanism of action of acetylcholinesterase inhibitors (AChEIs) is to block cholinesterase enzymes (AChE and butyrylcholinesterase (BChE)) from degrading ACh, resulting in increased levels of ACh in the synaptic gap [45]. The FDA-approved AChE inhibitors are donepezil, rivastigmine, and galantamine. AChE inhibitors can cause adverse effects like nausea, weight loss, and diarrhea. Donepezil can also cause insomnia, vomiting, anorexia, and asthenia. The adverse effects of rivastigmine are tremors, blurred vision, and confusion. Galantamine may lead to urinary retention and sinus bradycardia. Treatment with all of the AChE inhibitors requires monitoring for cardiac side effects [46].

An NMDAR antagonist prevents excessive activation of the NMDA glutamate receptor and thus Ca^2+^ influx and restores its normal activity. Memantine is an example of an N-methyl-D-aspartate (NMDA) receptor antagonist, a subtype of the glutamate receptor. It is commonly used in combination with acetylcholinesterase inhibitors to treat Alzheimer’s dementia. The purpose of memantine is to slow neurotoxicity associated with neurodegenerative diseases. Memantine is a non-competitive, low-affinity open-channel blocker that blocks the extrasynaptic NMDA receptor (NMDAR), which selectively enters the receptor-associated ion channel in the open state. As a result, the drug prevents the disruption of normal synaptic transmission, protecting against further damage caused by excitotoxicity-induced neuronal cell death [47]. Despite the fact that memantine is a well-tolerated drug, it can cause dizziness, falls, agitation, headache, diarrhea, and influenza-like symptoms [46]. The above-mentioned drugs are only effective in treating the symptoms of AD but do not cure or prevent the disease [48].

An atypical antipsychotic drug, called brexpiprazole, also helps with AD. Brexpiprazole acts on serotonergic, noradrenergic, and dopaminergic neurotransmitter systems that are involved in the neurochemistry of arousal in AD. Two previous randomized clinical trials have suggested that brexpiprazole, 2 mg, may be safe, works effectively, and is well tolerated in patients with agitation in Alzheimer’s-type dementia. The purpose of this clinical trial was to confirm the efficacy, safety, and tolerability of brexiprazole [49]. In May 2023, brexpiprazole was approved by the FDA for use in the treatment of agitation associated with dementia. Adverse effects of the treatment with this drug are headache, nasopharyngitis, insomnia, urinary tract infection, and dizziness. In addition, patients treated with brexpiprazole require monitoring against dystonia, neutropenia, agranulocytosis, akathisia, and cognitive impairment. Interestingly, the odds of mortality are higher compared to a placebo.

Another FDA-approved drug for AD is Suvorexant. It is an orexin receptor antagonist which is used to treat sleep disturbances in individuals with AD. Orexin is a neurotransmitter responsible for maintaining a state of wakefulness. Suvorexant, by blocking the action of orexin receptors, helps to initiate and maintain sleep. As the most common adverse effect is somnolence, it can also cause xerostomia, headache, dizziness, diarrhea, upper respiratory tract infection, dyspepsia, and peripheral edema [46].

A promising drug that treats the cause of AD is lecanemab. It is an antibody against soluble amyloid-beta protofibrils, which is administered by intravenous infusion every 2 weeks. Lecanemab reduced markers of amyloid in early Alzheimer’s disease and resulted in moderately less decline in measures of cognition and function than a placebo at 18 months but was associated with adverse events [50]. The Food and Drug Administration (FDA) approved lecanemab under an accelerated approval pathway followed by traditional approval in July 2023. Since then, there has been remarkable enthusiasm for lecanemab and renewed interest in this class of drugs (the FDA approved aducanumab in 2021) [51].

Donanemab is a promising drug that is in phase 3 in a few trials. It is a humanized monoclonal antibody that recognizes Aβ, an aggregated form of Aβ in amyloid plaques and bound up to one-third of plaques [52]. Compared to placebo, donanemab slowed down the progression of AD by over 30% [53].

It was observed that despite the positive outcome of AD treatment with immunotherapy, it can cause amyloid-related imaging abnormalities, mainly with edema [54]. The effects and adverse effects of described drugs are summed up in Table 1.

Studies have shown that physical activity plays a significant role in the prevention of AD. It benefits brain health and reduces AD progression because it activates cerebral vascularization. Moreover, it reduces inflammation by decreasing Aβ production, resulting in improved cognitive function in the elderly. Intellectual activity, a Mediterranean diet (MD), and higher education can reduce AD progression and memory loss and increase cognitive function [55].

## 5. Main Facts about Dendrimers

Dendrimers are covalently bonded globular nanostructures evocative of a tree 10 to 100 nm in size [56]. They were discovered in 1985 by Professor Donald Tomalia [57]. The etymology of the word “dendrimer” came from the Greek words “dendritic”, meaning tree-like, and “meros”, meaning part [56].

The structure of dendrimer compounds consists of three main components—core, branching units, and terminal groups [57], as shown in Figure 1. They demonstrate an extraordinary molecular architecture aggregating a core of the molecule from which the branches diverge and whose endings create the periphery of the molecule. The inside of the dendrimer consists of cavities that are becoming more compelling in the more advanced generation materials [58]. The branches themselves are responsible for the growth and size of dendrimers [59]. All of these elements are responsible for physicochemical attributes, for example, their functionality, charge of the surface, hydrophilicity, and conformational flexibility. These parts could be adjusted through proper chemical handling. Thus, many kinds of dendrimers have now been described by the kind of intended application [58]. Generally, dendrimers are created by repeated chemical reactions in an ongoing pattern, leading to the synthesis of new layers called generations [60].

The creation of dendrimers is divided into two separate strategies—divergent and convergent. The first method used to create dendrimers is divergent synthesis. At first, the core of dendrimers is called G0-generation zero. The addition of branches to the core until the required size is called a divergent strategy [59]. The core presents functional groups that interact with monomers, which results in generating G1 dendrimers. Each repeating branch from the core is the next generation—for example, G2, G3, G4, etc. [57]. That method is used especially for creating PAMAM-very precise and small nanomolecules [4]. PAMAMs’ diameter range from 1 nm to 14 nm, which corresponds with their generation (from G0 to G10). They are the most commonly studied family of dendrimers [61]. PAMAM-polyamidoamines have an impact on the mitophagy of microglia. Because of that, their influence on neuroprotection is considered [62]. Unfortunately, the divergent method is effectively longer than the convergent method because of the great number of reactive sites needed. However, this method can develop dendrimers during every sequence of adding branches when a convergence strategy is needed to wait for the final generation [59].

The second method of developing dendrimers, which was invented later, is the strategy of convergence. It was developed by Hawker and Frechet, and the general idea of this process is that the first branches are linked to each other until they reach the acquired size and then are connected to the core. 3,5-dihydroxybenzyl alcohol was the first factor in that process—undergoing reaction with the first-generation benzyl bromide and creating G1. Carbon tetramide and triphenyl phospine activation created G2; cycles of coupling and bromination led to higher generations. Finally, brominated dendrons were linked to the core consisting of 1,1,1-tris(4′-hydroxyphenyl)ethane. The lower amount of reagents, higher purification, and lower risk of side reactions are advantages of the method [57]. Undoubtedly, the advantage of convergence is a shorter time of reaction and easier purification of the product from the reactants [59].

Kawaguchi et al. described phenylacetylene dendrimers-phenyl monomer with three branches: a triazene focal point and two orthogonal trimethylsilyl-protected alkynes. These alkynes are required for stable division of the functional groups in the opposite direction. The method consists of selective deprotection of branched monomer end groups to create a convergent monomer. What is also important is that if the triazene focal point were deprotected, it would create a divergent monomer. It is called “double exponential growth” and connects both divergence and convergence synthesis of dendrimers. The specificity of coupling, higher purification, and efficiency are considered the upsides of the process.

The next described method of dendrimer synthesis, “hypercores” and “branched monomers” growth, uses oligomeric molecules synthesized earlier. These molecules are linked to hypercore with a small amount of diluent, making the purification easier [57].

“Onion peel” dendrimers are layered structures created by using cyclotriphosphazenes as the first layer and synthesized by ongoing reactions connecting alkynes and azides [63]. If the core is chosen, the next step of synthesis is photolytic thiolene coupling or thiolyne coupling, which creates active peripheral functions. Afterward, esterification or amidation is required for new layers with the complementary focal function. Finally, bioconjugation with copper-catalyzed azide-alkyne cycloaddition (CuAAc) is performed. The low risk of toxicity when using this method is suitable for the biological applications described later in this work [64].

PAMAM synthesis requires ammonia as a core, followed by three methyl acrylate groups that attach an ethylenediamine - structure called G1 dendrimer. Later, each closing nitrogen of ethylenediamine attaches two more methyl acrylates, creating G2 dendrimers. And so, an ongoing and repeating reaction leads to developing the final necessary dendrimer. PAMAM provides better treatment of HER2-positive breast cancer -and is used as a compound of trastuzumab [57], which allows for better bioavailability and increases pharmacokinetics [65]. A decrease in the side effects of therapy is described.

PAMAM can be purposely modified by changing its functional groups, creating structures that can be hydrophilic or hydrophobic. This process helps the drugs to improve their adaptation to homeostasis fluctuations, such as pH and membrane potential. Modern medicine uses dendrimers as drug stabilizers and dissolving factors. Dendrimers can be modified so their termini function as ionic or covalent bonds [58]. Dendrimers enter the cells via direct penetration or endocytosis, and when released from endosomes, they migrate to lysosomes [66], which is presented in Figure 2. PAMAM dendrimers are resistant to temperatures higher than ~200 °C and freezing. There were no notable examples of aggregation or denaturation during the defrosting of these molecules [67].

PAMAM’s biological impact on the expression of genes was examined. As a vector, PAMAM increases transfection effectiveness by even 40% in primary cortical neurons. During research on rats’ brains, ammonium-terminated carbosilane dendrimer successfully led to the delivery of siRNA to cortical neurons, which helped to decrease the number of proteins by eight-tenths [68].

The effectiveness of that process increases the bioavailability of drugs. Researchers have studied concentrations of memantine administered to rats in three forms: only memantine; memantine linked to dendrimers; and, the most effective, memantine linked to dendrimers conjugated to lactoferrin, ~2.17 times more potent than without lactoferrin [69].

The main disadvantage of using dendrimers is their cytotoxicity because of adding cationic polymers, which interfere with the negative membrane potential, leading to the dissolution of cells. The way to lower their undesirable impact on cells is using lower generations of dendrimers because then the number of positive charges is significantly reduced, and toxicity is limited [70].

## 6. Dendrimers in the Treatment of Alzheimer’s Disease

Nowadays, Alzheimer’s disease therapies are focused on improving cognitive functioning and have no causal treatment. This may be changed by using dendrimers, which are shown to be a drug-delivery system (for example, they can carry carbamazepine and tacrine to improve their bioavailability and mitochondrial targeting) or as a drug per se. Some studies show that dendrimers present abilities to inhibit amyloid formation and anti-tau properties.

The presence of amyloid fibrils is observed in several neurodegenerative disorders, including Alzheimer’s disease. Studies have shown that dendrimers may inhibit amyloid formation in at least two ways. The first one is blocking fibril growth by unspecifically binding to the fibril ends. Microscopic observations revealed that low-generation dendrimers (PAMAM G3) made fibrils more curved and organized in clumps. Higher generations, such as PAMAM G5, interact with monomeric fibrils. Microscopic observations detect fewer fibrils after interacting with those dendrimers. The second strategy is to break already existing fibrils. One study showed that the higher the dendrimer generation is, the more effective they are in breaking already existing fibrils [71].

Neurofibrillary tangles (NFTs) are abnormal filaments built by tau proteins that can be accumulated in neural perikaryal axons, dendrites, and cytoplasm. This accumulation leads to a loss of cytoskeletal microtubules and tubulin-associated proteins. In AD patients’ brains, hyperphosphorylated tau is the major component of NFTs, and as they form extracellular tangles, they result in neuronal loss [45]. Cationic phosphorus dendrimers (CPD) have shown anti-tau properties as they induce the aggregation of tau into amorphous structures instead of filamentous ones [72]. The effect of tau aggregation inhibition by G3 and G4 CPD depends on the dendrimer/peptide ratio. A high ratio (1.5) inhibits tau aggregation in vitro, while a low ratio has no effect at all. The ability to inhibit tau by G3 CPD comes from the induction of tau aggregation into amorphous structures, and G4 CPD shortens their fibrillar structures. [73]. These dendrimers also demonstrate the ability to modulate immune response and weak antioxidant properties without disrupting the work of AChE inhibitors [72]. They also inhibit acetylcholinesterase activity themselves [74]. A schematic representation of the dendrimers’ function is shown in Figure 3.

There is more and more information about how mitochondrial dysfunction among beta-amyloid and hyperphosphorylated tau proteins are driving forces in the development of Alzheimer’s disease [75]. As the importance of mitochondrial dysfunction as a part of AD pathogenesis increases, aiming at mitochondrial dysfunction as a therapeutic target seems to be reasonable [76]. Drug delivery to mitochondria is difficult due to poor transport across biological barriers [77], such as the highly negatively polarized inner mitochondrial membrane [78], which blocks the charged and polar molecules from entering the mitochondrial matrix. Hydroxyl-terminated PAMAM dendrimers are neutral, which makes them potential drug carriers because of their ability to cross the impaired blood–brain barrier. They accumulate in inflammation areas in activated microglia with no tendency to accumulate in healthy brain tissue. Positively charged PAMAM dendrimers interact with cellular components such as the nucleus, cell membrane, or endosomes. Their ability to cellular internalization makes them potential carriers for cellular and organelle delivery [58]. Bielski et al. investigated how the conjugation of triphenylphosphonium cation (TPP) to the G4 PAMAM dendrimer affects mitochondrial internalization [79]. TTP demonstrates a large hydrophobic surface, so their concentration increases a hundredfold in the mitochondria in comparison to the cytoplasm. This property is used by numerous therapeutic agents to target mitochondria [58]. This study consisted of different TTP dendrimer surface densities. Low densities of TTP increased cellular internalization and mitochondrial targeting. At a greater density, a further increase was observed, but unfortunately, the nanocarriers started to show evident cytotoxicity. The investigators observed that adding a PEG linker between dendrimer and TTP decreases the toxicity without affecting the mitochondrial targeting [79].

The other considered utility of dendrimers in AD treatment is to apply them as drug carriers. Carbamazepine (CBZ), which is known as a mood-stabilizing and anticonvulsant drug, has been found to have an autophagy-enhancement effect [80]. Used in mice models, carbamazepine showed the potential to reduce the amount of amyloid plaques in the hippocampus [81]. In vitro, in cell models, it reduces toxic effects and the accumulation of aggregation-prone proteins, and in vivo, it performs protective effects against neurodegeneration. Since CBZ shows incomplete drug bioavailability due to poor solubility in water as well as slow and irregular gastrointestinal absorption, it has to be delivered to the brain by some kind of drug-delivery system (DDS). Dendrimers 4.0 and 4.5 PAMAM are characterized by a small and controlled size, and high water solubility may be a drug-delivery system for carbamazepine. Previous studies showed that PAMAM dendrimers used as DDS for risperidone and sulfadiazine enhance their arrival in the brain and the potency of their effect [80].

Currently, the treatment of Alzheimer’s disease is based on acetylcholinesterase (AChE) inhibitors such as rivastigmine, donepezil, and galantamine. Tacrine (TAC) was the first AChE inhibitor registered by the FDA for AD treatment, and it is the most effective AChE inhibitor [82]. However, its clinical application is limited due to its hepatotoxicity and peripheral cholinergic side effects. There is a tendency to administer TAC by the nasal or transdermal route to reduce these side effects by avoiding the hepatic first-pass effect [83]. Using PAMAM G4.0 and G4.5 dendrimers as a drug-delivery system for tacrine has been investigated. The tacrine/dendrimers mixtures were found to be equally effective as tacrine itself, there were no cytotoxic or hemolytic side effects, and the hepatotoxicity was reduced [58].

Prussian blue/polyamidoamine (PAMAM) dendrimer/Angiopep-2 (PPA) nanoparticles are promising treatments for patients who already have AD symptoms. Prussian blue has antioxidant effects through ROS removal. This suggests the possible neuroprotective effects for AD due to treating mitochondrial oxidative stress-induced damage. As Prussian blue poorly crosses the blood–brain barrier (BBB), its incorporation with PAMAM dendrimers may increase the BBB transport efficacy and circulation lifetime [62].

Beta-site amyloid precursor protein cleaving enzyme-1 (BACE1) initiates Aβ monomer production. It cuts amyloid precursor protein (APP) at the β-secretase site. Zhang et al. created an NL4-ApoA-I-siRNA-dendrimer to transport siRNA into neuronal tissue. This complex suppresses BACE1, reducing Aβ generation [84].

Polypropylene Imine (PPI) dendrimers possess a highly branched structure and can be functionalized with various bioactive molecules, such as antioxidants and metal chelators, to combat oxidative stress and metal ion dysregulation implicated in Alzheimer’s disease. PPI dendrimers may help preserve neuronal function and delay disease progression by reducing oxidative damage and mitigating the toxic effects of excess metal ions in the brain [85].

Furthermore, Carbosilane dendrimers, based on silicon–carbon linkages, offer unique properties, such as stability and biocompatibility. These dendrimers can carry multiple therapeutic agents, including anti-inflammatory drugs and neuroprotective peptides. Their ability to cross the blood–brain barrier and deliver these agents directly to the central nervous system makes them promising candidates for Alzheimer’s therapy. Carbosilane dendrimers can also be designed to interact with specific cellular receptors, enhancing their targeting efficiency and therapeutic efficacy in mitigating the neurodegenerative processes associated with Alzheimer’s disease [86]. These advancements highlight the potential of dendrimer-based nanotherapeutics in offering a novel approach to Alzheimer’s disease management. 

The mechanisms described above are shown in Table 2.

## 7. Dendrimers in Treatment of Other Diseases

Dendrimers may be also used for the treatment of other neurodegenerative diseases. One example may be dendrimers used for drug delivery in Parkinson’s disease (PD) [87]. Dendrimers may be used as a new nanotechnology invention to provide tools to stabilize α-syn fibers and prevent the formation of α-syn aggregates [88]. Interestingly, dendrimers are said to inhibit the aggregation of insoluble forms of prion peptides and amyloid. Moreover, dendrimers may facilitate its degradation [89,90,91].

Dendrimers may be used not only in neurodegenerative disease treatment. These molecules may serve as drugs for cancer. Various cancer therapies have been proposed, such as cytokine-based immunotherapies [92], cancer vaccines [93], and monoclonal antibodies for targeted therapy [94]. In cancer therapy, dendrimers such as polyamidoamine (PAMAM) are utilized to deliver chemotherapy drugs directly to cancer cells, enhancing the drugs’ effectiveness while minimizing side effects [95]. Interestingly, dendrimers have been reported as a drug for various infectious diseases [96]. Many studies claim that some dendrimers exhibit activities against many pathogens [97], even for SARS-CoV2 virus [98].

Dendrimers are gaining recognition as potential therapeutic agents for a variety of diseases thanks to their unique, highly branched structure and adaptability. In the treatment of viral infections, carbosilane dendrimers have shown promise, particularly against HIV, by blocking the virus’s ability to enter and replicate within host cells [99]. For genetic disorders, dendrimers can be employed in gene therapy to deliver genetic material to specific cells, thereby addressing the root cause of the mutation [100]. Additionally, dendrimers are being investigated for their role in cardiovascular disease treatments, such as delivering drugs that prevent the re-narrowing of arteries after procedures like angioplasty [101]. The ability to attach multiple therapeutic agents to their surfaces makes dendrimers a versatile and powerful tool for treating a diverse array of diseases.

## 8. Conclusions

The number of people affected by Alzheimer’s disease will continue to grow. Patients often require daily care, which is a challenge for families, both emotional and financial, as the condition of sufferers often deteriorates. Dendrimers can have a major impact on the course and treatment of Alzheimer’s disease (AD), which is the most common neurodegenerative disease and has posed challenges to improving its treatment. This research presents a study that shows that dendrimers can inhibit amyloid production and have anti-tau properties. Dendrimers are constantly being studied as a drug-delivery system, for example, carbamazepine (CBZ) or tacrine, or as a drug in its own right. The tacrine/dendrimer mixtures were found to be equally effective as tacrine itself. Dendrimers may also be used to transport siRNA into neuronal tissue. It has been proven that dendrimers can be potential nanocarriers and enhance mitochondria targeting. Moreover, positively charged PAMAM dendrimers interact with cellular components such as the nucleus, cell membrane, or endosomes. They have the capacity for cellular internalization, which makes them potential carriers for cellular and organelle delivery.

## Figures and Tables

**Figure 1 biomedicines-12-01899-f001:**
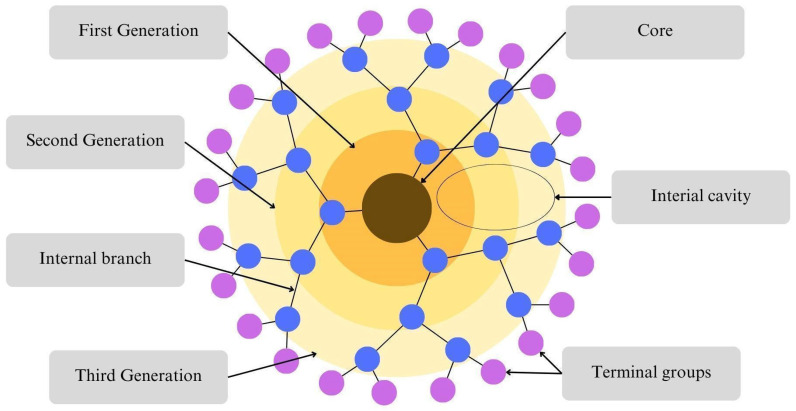
Structure of a dendrimer.

**Figure 2 biomedicines-12-01899-f002:**
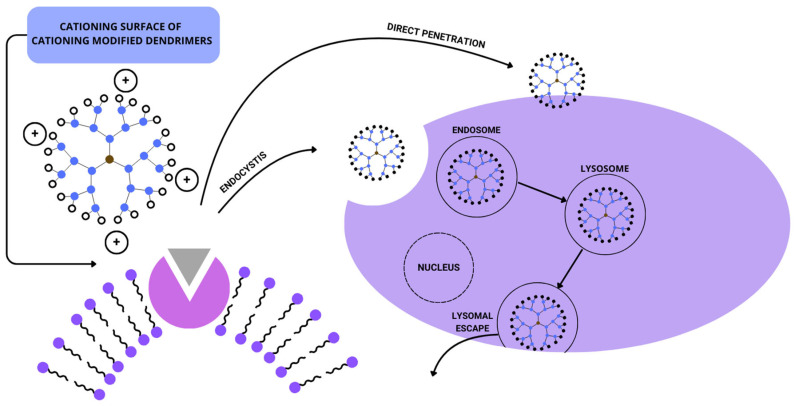
Schematic representation of potential intracellular organelles targeted by dendrimers. Partially adapted from [66].

**Figure 3 biomedicines-12-01899-f003:**
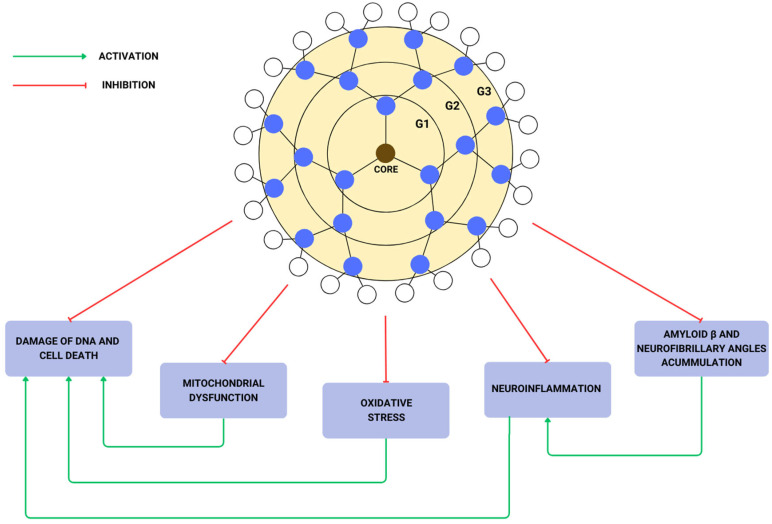
Dendrimer structure and its influence on neuron cell.

**Table 1 biomedicines-12-01899-t001:** The effects and adverse effects of the drugs.

Structure	Drug	Effects	Adverse Effects
cholinesterase enzyme inhibitor	Donepezil	Increases level of acetylcholinesterase in the synaptic gap [45]	nausea, weight loss, diarrhea, insomnia, vomiting, anorexia, asthenia [46]
	Rivastigmine		nausea, weight loss, diarrhea, tremors, blurred vision, confusion [46]
	Galantamine		nausea, weight loss, diarrhea, urinary retention, sinus bradycardia [46]
N-methyl-d-aspartate antagonist	Memantine	Slow neurotoxicity associated with neurodegenerative diseases [47]	dizziness, falls, agitation, headache, diarrhea, influenza-like symptoms [46]
Atypical antipsychotic drug	Brexipiprazole	Treatment of agitation associated with AD [46]	headache, nasopharyngitis, insomnia, urinary tract infection, dizziness [46]
Orexin receptor antagonist	Suvorexant	Treatment of sleep disturbances associated with AD [46]	somnolence, xerostomia, headache, dizziness, diarrhea, upper respiratory tract infection, dyspepsia, peripheral edema [46]
Monoclonal antibodies against amyloid-beta	Lecanemab	Binding amyloid-beta [52]	amyloid-related imaging abnormalities, mainly with edema [54]
	Aducanumab		
	Donanemab		

**Table 2 biomedicines-12-01899-t002:** Explanation of possible dendrimer uses in Alzheimer’s disease treatment.

	Mechanism
Amyloid fibrils	Dendrimers may inhibit amyloid formation by blocking their growth or breaking already existing ones [71]
Neurofibrillary tangles	Cationic phosphorus dendrimers induce aggregation of tau into amorphous structures instead of filamentous ones [72]
Mitochondrial dysfunction	Dendrimers may be potential nanocarriers and increase mitochondria targeting [58]
Drug-delivery system	Carbamazepine, due to its autophagy-enhancement effect, has the potential to reduce the amount of amyloid plaques in the hippocampus [81]
	Tacrine is the most effective AChE inhibitor. Using dendrimers as DDS may reduce side effects [58]
	Prussian blue treats mitochondrial oxidative stress-induced damage [62]
	NL4-ApoA-I-siRNA suppresses BACE1, reducing Aβ generation [84]
	Polypropylene Imine (PPI) dendrimers can be functionalized with various bioactive molecules, such as antioxidants and metal chelators, to combat oxidative stress and metal ion dysregulation [85]
	Carbosilane dendrimers carry multiple therapeutic agents, including anti-inflammatory drugs and neuroprotective peptides directly to the central nervous system [86]

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
