# Peer review of "Dendrimers—Novel Therapeutic Approaches for Alzheimer’s Disease"

_biomedicines, 2024, doi:10.3390/biomedicines12081899_

Round 1

Reviewer 1 Report (Previous Reviewer 1)

Comments and Suggestions for Authors

The present manuscript has revised a lot according to the Reviewers’ comments and suggestions. The following points should be checked again before it can be accepted for publication.

**L10: Abstract: It is better to highlight the aim and/or the specific content of this review article in the Abstract.

*L21-22: Dendrimers may also inhibit amyloid formation by blocking their growth or breaking already existing ones. -> repeated statement (vs. L15-17)

L31: AD is -> Alzheimer’s Disease (AD) is

*L59: Alzheimer's uses?

L116:  radical anion (O2–), hydrogen peroxide (H2O2), hydroxyl radical (HO–)-> note the subscript or superscript

**L132: estrogens exacerbate neuroinflammation?-> loss of estrogens exacerbate neuroinflammation

*L133: why women have an increased risk of AD -> why older women have an increased risk of AD

L146: Ca2+ influx -> Ca2+ influx (superscript)

**L172: aripiprazole? Ref 49 is for Brexpiprazole.

**L366-369: Ref. 78 is a review paper: Unnisa A, Greig NH, Kamal MA. Nanotechnology-based gene therapy as a credible tool in the treatment of Alzheimer's disease. Neural Regen Res 2023 Oct;18(10):2127-2133. doi: 10.4103/1673-5374.369096, while the cited Zhang’s paper in this paper (L367) should be: Zhang C, Gu Z, Shen L, Liu X, Lin H. In vivo Evaluation and Alzheimer's Disease Treatment Outcome of siRNA Loaded Dual Targeting Drug Delivery System.

Curr Pharm Biotechnol. 2019;20(1):56-62. doi: 10.2174/1389201020666190204141046. PMID: 30727887

*L383: Table 1: It is better also to give the number of cited references

*L384: For the conclusion, it should add the potential use as SiRNA drug delivery system

*Inconsistent writing format for the References:

R6: Handbook of clinical neurology. -> Handbook of Clinical Neurology.

R14, 18, 19, 21, 24, 26, 28, 33, 34, 37, 42, 43, 45, 47, 49, 52, 53, 54, 57, 59, 60, 61, 62, 63, 64, 66, 67, 70, 71, 73, 75, 77, 79, 80, 83, 86, 88, 89: title (all capital prefix) -> capital prefix only on the first word to keep the consistent writing type with the other references.

*R84: no page number

*R86: Volume 15:

Comments on the Quality of English Language

Minor editing of English language required

Author Response

Thank you very much for your comments. Detailed responses are in the attached file.

Reviewer 2 Report (Previous Reviewer 2)

Comments and Suggestions for Authors

This resubmitted manuscript presents the application of dendrimers as potent therapeutic strategies for Alzheimer’s disease (AD). Although the authors have revised the manuscript based on the reviewers’ comments and summarized many previously published papers, this manuscript does not offer new findings compared to other review papers covering dendrimers and/or AD. Therefore, the manuscript is not suitable for the publication in Biomedicines. Detailed comments are provided below.

1.      What is the different from previous review papers? Unfortunately, there are already many similar works on this topic, and a search of “PubMed” using the keyword as “Dendrimers and Alzheimer’s disease” yields 28 review papers published within last 5 years (since 2019). Therefore, the manuscript should offer a unique perspective or recent findings which could be distinguished from previous review papers.

-          The authors answered that this manuscript is focusing on the synthesis methods and possible use of dendrimers in treatment of other diseases. However, the detailed synthesis methods are missing in this manuscript. In addition, this manuscript does not cover the other diseases noticeably.

2.      The authors should fix the chemical formula. Make them in correct way to present, especially subscript (for number of atoms in molecule) and superscript (for charges). They are not corrected at all.

3.      The authors added more recent references for this resubmitted review manuscript. Although the authors corrected the format of references, still there are some missing information of the references. All references should contain detailed information.

4.      It would be better to present i) the structure of FDA-approved drugs or promising drugs with their effects/side effects, ii) previous targets for developing treatments of AD, iii) examples of the dendrimers which can reduce the risk of onset and/or progression of AD. The authors added 2 new figures, however, I do not thinks they are covering this comment.

Author Response

Thank you very much for your comments. Detailed responses are in the attached file.

Round 2

Reviewer 2 Report (Previous Reviewer 2)

Comments and Suggestions for Authors

The authors have revised and improved the quality of this manuscript based on the reviewers' comments (including my previous comments).

Now, all my concerns are cleared.

This revised manuscript is suitable for publicaion.

This manuscript is a resubmission of an earlier submission. The following is a list of the peer review reports and author responses from that submission.

Round 1

Reviewer 1 Report

Comments and Suggestions for Authors

This paper reviewed the use of Dendrimers in the treatment of Alzheimer’s disease (AD). The potential effects of Dendrimers for AD include inhibiting amyloid formation, anti-tau properties, as nanocarriers to increase mitochondria targeting, as carriers for cellular and organelle delivery, and as drug delivery system for carbamazepine, Tacrine, and others. There are some concerns as listed in the following:

L44: Alzheimer’s disease -> AD

**L102: cytochrome I -> mitochondrial complex I

**L103: another cytochrome inhibitor -> another mitochondrial complex I inhibitor

*L109: mitochondrial proteasis -> mitochondrial proteostasis

*L119-L135: The cited reference number was disorder -> reassign the reference number

*L130-132: Interestingly, estrogens exacerbate neuroinflammation, this may be also the reason why women have increased risk of AD [36]. Administration of estrogen agonists protects the brain, limiting neuroinflammation [37]. -> estrogens exacerbate neuroinflammation vs. estrogen agonists protects the brain, limiting neuroinflammation??

L140: Ach-producing cells -> ACh-producing cells

L144: NMDAR glutamate receptor

L144: Ca2+ influx -> Ca2+ influx

**L241: 6. Dendrimers in Treatment of Alzheimer’s Disease -> Consider cite these two papers [Gang Zhong et al. Blood-brain barrier Permeable nanoparticles for Alzheimer's disease treatment by selective mitophagy of microglia. Biomaterials . 2022 Sep:288:121690. doi: 10.1016/j.biomaterials.2022.121690. Epub 2022 Aug 12.] and [Aziz Unnisa et al. Nanotechnology-based gene therapy as a credible tool in the treatment of Alzheimer's disease. Neural Regen Res 2023 Oct;18(10):2127-2133. doi: 10.4103/1673-5374.369096] into this section. Then, modify the Table 1 if they are cited.

L308: ACheE inhibitor -> AChE inhibitor

*Inconsistent writing format for the References

L353: R6: no journal?

L368: R14: title (all capital prefix)

L377: R18: title (all capital prefix)

L379: R19: title (all capital prefix)

L382: R21: title (all capital prefix)

L388: R24: title (all capital prefix)

L392: R26: title (all capital prefix)

L396: R28: title (all capital prefix)

L406: R33: title (all capital prefix)

L419: R40: title (all capital prefix)

L421: R41: title (all capital prefix)

Author Response

Thank you for your helpful comments. In attached file are our responds.

Reviewer 2 Report

Comments and Suggestions for Authors

This manuscript presents the application of dendrimers as potent therapeutic strategies for Alzheimer’s disease (AD). While the authors have summarized many previously published papers, this manuscript does not offer new findings compared to other review papers covering dendrimers and/or AD. Therefore, the manuscript is not suitable for the publication in Biomedicines. Detailed comments are provided below.

1.      What is the different from previous review papers? Unfortunately, there are already many similar works on this topic, and a search of “PubMed” using the keyword as “Dendrimers and Alzheimer’s disease” yields 26 review papers published within last 5 years (since 2019) and 10 review papers have been published in 2023 and 2024. Therefore, the manuscript should offer a unique perspective or recent findings which could be distinguished from previous review papers.

2.      This review manuscript contains 73 references, but only 28 references (about 38%) have been published from the last 5 years (since 2019). Since many similar papers are existed, the authors should focus on the latest findings and provide a comprehensive review of the most recent researches. Also, the authors should ensure that all references contain detailed information, such as journal name, year, volume #, or page #.

3.      In general, this manuscript needs more figures and tables to summarize the contents. For example, i) the structure of FDA-approved drugs or promising drugs with their effects/side effects, ii) previous targets for developing treatments of AD, iii) examples of the dendrimers which can reduce the risk of onset and/or progression of AD.

4.      What is “XX century”? Does the authors mean “20 century”? Also, please fix some grammatical errors.

5.      Please fix the chemical formula. Make them in correct way to present, especially subscript (for number of atoms in molecule) and superscript (for charges).

6.      It is not clear why the authors mentioned the cytochrome oxidase inhibitors in section 3.

7.      The Figure 1 looks very similar to the Figure 2 and 3 in the following manuscript: 10.3390/pharmaceutics15041054.

Comments on the Quality of English Language

Please fix some grammatical errors.

Please fix the chemical formula. Make them in correct way to present, especially subscript (for number of atoms in molecule) and superscript (for charges).

Author Response

(The authors gave the same response as above.)

Reviewer 3 Report

Comments and Suggestions for Authors

In the manuscript titled “Dendrimers - Novel Therapeutic Approaches for Alzheimer’s 2 Disease” discusses Dendrimers which are promising nanostructures for innovative Alzheimer's disease (AD) treatment, potentially offering both drug delivery systems and active therapeutic agents. They can inhibit amyloid formation by blocking fibril growth and breaking existing fibrils and have anti-tau properties that prevent neurofibrillary tangles, which are detrimental in AD. Additionally, dendrimers can enhance drug delivery to mitochondria, overcoming biological barriers. Their use in delivering drugs like Carbamazepine or Tacrine shows potential for more effective AD therapies. Overall, dendrimers represent a significant advancement in targeting the underlying causes of AD.

Despite the significant effort,  this manuscript requires editing to enhance its quality.

1.  The manuscript does not provide a thorough comparison between dendrimers and other nanoparticle-based methods for AD treatment. Include a section that evaluates the advantages and disadvantages of dendrimers over other nanoparticles, such as liposomes, polymeric nanoparticles, and gold nanoparticles. Discuss aspects like biocompatibility, targeting efficiency, drug loading capacity, and potential side effects.

2.  The manuscript lacks a generalized method for dendrimer synthesis and their evolution over time. Add a detailed overview of the synthesis methods for dendrimers, emphasizing improvements and optimizations that have been developed. Highlight how these advancements have enhanced their efficacy and safety in AD treatment.

3. The manuscript focuses solely on AD and does not explore the potential application of dendrimers in treating other diseases. Expand the discussion to include the potential use of dendrimers in other neurodegenerative diseases (e.g., Parkinson’s disease), cancer, and infectious diseases. Provide examples of studies that have investigated dendrimers in these contexts.

4.  The manuscript lacks visual aids to illustrate dendrimer delivery and their mode of action. Incorporate detailed schematic diagrams that show the structure of dendrimers in relation to their mechanism of drug delivery, and their interaction with amyloid fibrils and tau proteins. Visual representations of how dendrimers target mitochondria would also enhance understanding.

Author Response

(The authors gave the same response as above.)
